# ProCPU Is Expressed by (Primary) Human Monocytes and Macrophages and Expression Differs between States of Differentiation and Activation

**DOI:** 10.3390/ijms24043725

**Published:** 2023-02-13

**Authors:** Karen Claesen, Joni De Loose, Pieter Van Wielendaele, Emilie De bruyn, Yani Sim, Sofie Thys, Ingrid De Meester, Dirk Hendriks

**Affiliations:** 1Laboratory of Medical Biochemistry, Department of Pharmaceutical Sciences, University of Antwerp, 2610 Wilrijk, Belgium; 2Laboratory of Cell Biology and Histology, Faculty of Pharmaceutical, Biomedical and Veterinary Sciences, University of Antwerp, 2610 Wilrijk, Belgium

**Keywords:** carboxypeptidase B2, carboxypeptidase U, gene expression, monocyte/macrophage, thrombin-activatable fibrinolysis inhibitor

## Abstract

Carboxypeptidase U (CPU, TAFIa, CPB2) is a potent attenuator of fibrinolysis that is mainly synthesized by the liver as its inactive precursor proCPU. Aside from its antifibrinolytic properties, evidence exists that CPU can modulate inflammation, thereby regulating communication between coagulation and inflammation. Monocytes and macrophages play a central role in inflammation and interact with coagulation mechanisms resulting in thrombus formation. The involvement of CPU and monocytes/macrophages in inflammation and thrombus formation, and a recent hypothesis that proCPU is expressed in monocytes/macrophages, prompted us to investigate human monocytes and macrophages as a potential source of proCPU. *CPB2* mRNA expression and the presence of proCPU/CPU protein were studied in THP-1, PMA-stimulated THP-1 cells and primary human monocytes, M-CSF-, IFN-γ/LPS-, and IL-4-stimulated-macrophages by RT-qPCR, Western blotting, enzyme activity measurements, and immunocytochemistry. *CPB2* mRNA and proCPU protein were detected in THP-1 and PMA-stimulated THP-1 cells as well as in primary monocytes and macrophages. Moreover, CPU was detected in the cell medium of all investigated cell types and it was demonstrated that proCPU can be activated into functionally active CPU in the in vitro cell culture environment. Comparison of *CPB2* mRNA expression and proCPU concentrations in the cell medium between the different cell types provided evidence that *CPB2* mRNA expression and proCPU secretion in monocytes and macrophages is related to the degree to which these cells are differentiated. Our results indicate that primary monocytes and macrophages express proCPU. This sheds new light on monocytes and macrophages as local proCPU sources.

## 1. Introduction

Monocytes and macrophages play a central role in the inflammatory response in atherosclerosis and at extra-vascular inflammatory sites [1,2,3]. In addition, these cells can interact with blood coagulation mechanisms, leading to thrombus formation or extravascular fibrin deposition [4,5]. Numerous macrophage subtypes have been identified, with IFN-γ/LPS- and IL-4-stimulated macrophages representing the opposite sites of the macrophage spectrum [2,6,7,8,9]. IFN-γ/LPS-stimulated macrophages (M1- or classically activated macrophages) are important producers of pro-inflammatory cytokines. IL-4-stimulated macrophages (M2- or alternatively activated macrophages) are producers of anti-inflammatory cytokines [2,10,11].

The intrinsically unstable carboxypeptidase U (CPU, TAFIa, CPB2) is a potent attenuator of fibrinolysis and a possible modulator of inflammation that is present in the circulation as its zymogen procarboxypeptidase U (proCPU, TAFI, proCPB2). Plasma proCPU mainly originates from the transcription of the *CPB2* gene in the liver [12,13,14]. However, other cell types have been identified as (potential) additional proCPU sources. A first non-hepatically derived pool of proCPU was found in the platelets and accounts for <0.1% of blood-derived proCPU. It is synthesized by megakaryocytes and released from the α-granules upon platelet activation [15]. *CPB2* mRNA was also detected in megakaryocytic cell lines (CHRF, Dami and MEG-01), primary endothelial cells (both HCAEC and HUVEC), and the human monocytic cell line THP-1 as well as in THP-1 cells differentiated into a macrophage-like phenotype and in peripheral blood mononuclear cells (PBMCs) [15,16,17,18,19]. In the promonocytic cell line U937, *CPB2* mRNA expression increased after treatment with dexamethasone or M-CSF [20]. ProCPU protein was detected in the lysate of differentiated and undifferentiated Dami and MEG-01 cells and the conditioned media of differentiated Dami and PMA-stimulated THP-1 cells [16].

*CPB2* mRNA was recently detected in PBMCs and it was hypothesized that the *CPB2* transcripts were derived from *CPB2* gene expression by monocytes and macrophages present in this cell fraction [16]. Since monocytes and macrophages provide a potential link between inflammation and thrombus formation [3], and the CPU system also plays a role in both systems, this is an interesting hypothesis. ProCPU expression has, however, not been studied separately in PBMC-derived monocytes. In addition, proCPU expression in macrophages has thus far only been studied in cell lines and not yet in primary cells that more closely mimic in vivo human physiology. Therefore, we investigated, for the first time, the expression of proCPU (on mRNA, protein and activity level) not only in the primary human monocytes, but also in different primary human macrophage subsets to gain more insights into these cells as potential sources of proCPU.

## 2. Results

### 2.1. CPB2 mRNA Is Detected in (Primary) Human Monocytes and Macrophages

Using the validated reverse transcriptase-polymerase chain reaction (RT-qPCR) assay (Appendix A), *CPB2* mRNA expression was studied in human monocyte and macrophage cell lines and primary cells and displayed relative to the expression of the reference genes selected for accurate normalization (ARPC1a, EMC7, and TBP; Appendix A). The human hepatocellular carcinoma cell line HepG2 was used as a positive control given that hepatocytes are the primary source of proCPU protein in plasma. RT-qPCR results showed that *CPB2* mRNA was present in all of the studied cell types, with the highest expression detected in HepG2 cells Figure 1. When comparing the *CPB2* mRNA abundance in the monocytic cell line THP-1 with primary human monocytes, the expression of the *CPB2* transcripts was significantly lower in the THP-1 cells compared to its primary cell counterpart. Although the magnitude of relative *CPB2* mRNA expression was different between the primary monocytes and THP-1 cells, a clear and significant decrease in *CPB2* mRNA expression was observed for both cell types when differentiating these cells into macrophages with M-CSF or PMA, respectively. Furthermore, activation of primary M-CSF macrophages with LPS/INF-ɣ or IL-4 resulted in differential *CPB2* gene expression: LPS/INF-ɣ activation gave rise to a slightly, but not significantly higher *CPB2* mRNA abundance compared to M-CSF macrophages, whereas IL-4 activation significantly lowered the expression of *CPB2* transcripts.

### 2.2. ProCPU and CPU Protein Are Present in (Primary) Human Monocyte and Macrophage Medium

The presence of the proCPU protein in the concentrated conditioned medium of human monocytes and macrophages was studied by Western blotting using two different polyclonal proCPU/CPU antibodies. ProCPU purified from plasma as well as CPU obtained after activating purified proCPU by the addition of thrombin-thrombomodulin were included as positive controls. Both antibodies against proCPU/CPU reacted with purified proCPU and CPU at a MW around 58 kDa and 35 kDa, respectively (Figure 2), corresponding with the previously reported data on human proCPU/CPU [21]. Using the sheep polyclonal proCPU antibody (PATAFI-S, Prolytix), a similar proCPU immunoreactive band was detected in all of the concentrated conditioned media samples, though the apparent MW was slightly lower compared to the purified proCPU (Figure 2A). With this antibody, no CPU band was observed in the media samples. The second Western blot showed a 35 kDa CPU band for all conditioned media, but this polyclonal antibody (CP17, Agrisera) did not react with proCPU in any of the media samples (Figure 2B).

An additional Western blot experiment was performed to gain more insights into the reactivity of both polyclonal antibodies against proCPU and CPU (Figure 3). PATAFI-S was found to better recognize proCPU, while CP17 showed a higher affinity toward CPU.

In order to identify that the 35 kDa protein band detected in the conditioned media with the CP17 antibody was truly CPU and to substantiate that proCPU can be activated into functionally active CPU in the in vitro cell environment, HepG2, THP-1, and PMA-stimulated THP-1 cells were cultured in the presence of 1 mM Bz-*o*-cyano-Phe-Arg, a specific CPU substrate. At different time points, the medium was collected and subjected to RP-HPLC following an in-house protocol of HPLC-assisted CPU activity measurement to investigate whether the substrate had been cleaved in the cellular environment [22]. As shown in Figure 4, the cleaved substrate (Bz-*o*-cyano-Phe) was detected in the medium of all of the tested cell types.

Lysate from all cell types including HepG2 was also subjected to Western blot analysis. Despite all our efforts, neither proCPU nor CPU could be detected by Western blotting in any of the cell lysates using the proCPU/CPU antibody. However, the presence of proCPU/CPU inside the different cell types was confirmed by immunocytochemistry (Appendix A).

### 2.3. ProCPU Concentration Measured in Medium of (Primary) Human Monocytes and Macrophages Is Related to Their State of Differentiation

ProCPU was measured in conditioned medium samples, with the highest levels seen in the HepG2 medium (Figure 5). The ProCPU concentration was similar in the medium of THP-1 cells and primary human monocytes and was significantly higher compared to the concentration the in medium of PMA-stimulated THP-1 cells and primary human M-CSF primed macrophages, respectively. Moreover, a slight but non-significant increase in proCPU concentration was observed after stimulation of the primary human M-CSF macrophages with IFN-γ and LPS, whereas IL-4 stimulation led to a further significant decrease in proCPU activity. In the cell lysates, the proCPU concentration was comparable in all of the studied cell types.

## 3. Discussion

The involvement of both CPU and monocytes/macrophages in inflammation and thrombus formation [3], and the hypothesis of Lin et al. that monocytes and macrophages are responsible for the *CPB2* mRNA expression of PBMCs [16], prompted us to investigate the expression of proCPU in (primary) human monocytes and different (primary) human macrophage subsets to gain more insights into these cells as potential sources of proCPU. For this research, we used the human monocytic cell line THP-1 and primary human monocytes and macrophages. The THP-1 cell line has been extensively used to study monocyte/macrophage function and biology, but suffers from the disadvantage that THP-1 cells differ genetically and phenotypically from primary monocytes. Primary cells such as PBMC-derived monocytes mimic the in vivo human physiology more closely, making it often a more relevant cell culture model [23,24]. Therefore, we included the primary human monocytes and macrophages derived from these monocytes. Moreover, this study was the first to examine proCPU expression separately in PBMC-derived human primary monocytes and macrophages and not in the PBMC-fraction as a whole.

After successfully validating a RT-qPCR assay to study *CPB2* mRNA expression, relative *CPB2* mRNA expression was determined in human monocyte- and macrophage cell lines and primary cells. Primary human monocytes displayed considerable *CPB2* mRNA expression while the expression decreased substantially after differentiating these cells into resting macrophages by the addition of M-CSF. Similar results were obtained in the THP-1 monocytic cell line and PMA-stimulated THP-1 cells. This is in line with the observations of Lin et al., although based on their results, we expected *CPB2* mRNA expression to be of similar magnitude in the primary human monocytes and THP-1 cells, but this was not the case [16]. Moreover, in databases containing the results of experiments in which mRNA was determined by microarray or single cell RNA sequencing mainly in mice, very low levels of *CPB2* mRNA were found in the monocytes and macrophages. Here, the abundance of *CPB2* transcripts was clearly higher in the primary cells. The higher expression observed here might be due to the fact that *CPB2* mRNA expression was studied specifically in monocytes and not in the whole PBMC fraction (of which monocytes make up 10–30%) as Lin et al. did. Furthermore, as mentioned earlier, cell lines and primary cells may genetically differ, and this might account (in part) for the difference in *CPB2* mRNA expression observed between the THP-1 cells and primary human monocytes. Moreover, it is difficult to quantitatively compare the RT-qPCR results obtained in this study and those of Lin et al. because here, the *CPB2* mRNA expression was expressed relative to a pool of stable reference genes, while Lin and co-workers made use of RNA standards for absolute quantification. Nevertheless, a clear trend between the expression of *CPB2* transcripts and the state of differentiation was observed for both the monocytic cell line and primary monocytes and macrophages.

Macrophages display remarkable plasticity and can change their physiology in response to environmental cues [25]. As a result, numerous macrophage subtypes with distinct functions have been identified [2,6,7,8,9,26]. In this context, we were also interested in whether the *CPB2* mRNA expression was different between resting M-CSF macrophages and so called M1- and M2-macrophages, representing the opposite sidesof the diverse macrophage spectrum. Primary M-CSF primed macrophages incubated in the presence of IFN-ɣ/LPS develop into pro-inflammatory macrophages, while IL-4 differentiates monocytes into anti-inflammatory macrophages. Interestingly, activation of primary human M-CSF macrophages resulted in differential *CPB2* mRNA expression: classical macrophage activation gave rise to a slightly, but not significantly higher *CPB2* mRNA abundance compared to the M-CSF primed macrophages, whereas alternative activation significantly lowered the expression of the *CPB2* transcripts. These observations further support that the expression of *CPB2* mRNA in monocytes and (activated) macrophages are related to the degree to which these cells are differentiated and/or activated.

In the concentrated conditioned medium of all of the studied cell types, the proCPU protein was detected using a polyclonal sheep anti-human proCPU antibody (PATAFI-S). Notably, the protein band appeared at a slightly different MW compared to the plasma purified proCPU (58 kDa). Due to the concentration of the conditioned medium samples, albumin (with a MW of 67 kDa) was present in a high concentration in these samples. It is possible that this high albumin concentration affects the electrophoretic transfer of proteins with a similar MW such as proCPU. This might explain why proCPU was detected on the Western blot at an apparently lower molecular weight than expected. With this antibody, no CPU was detected in the concentrated conditioned medium samples. In an attempt to detect proCPU or CPU protein in the cellular lysate samples, Western blotting was repeated on those samples using a second proCPU/CPU antibody and the same was carried out for the conditioned medium samples. To our surprise, this polyclonal rabbit anti-human proCPU/CPU antibody (CP17) visualized a protein band at a MW of around 35 kDa in all samples, exactly at the same level as the protein band of the purified CPU. Unexpectedly, no 58 kDa proCPU band was detected on the Western blot for any of the media samples using the CP17 antibody. A similar finding with the CP17 antibody was reported by Rylander et al., who suggested that denaturation with SDS may affect proCPU more than CPU, making proCPU more rigid and less detectable with the CP17 antibody. Comparing the reactivity of both the PATAFI-S and CP17 antibody toward purified plasma proCPU and CPU on the Western blot, the PATAFI-S antibody seemed to have a higher affinity toward proCPU, while CP17 reacted better with CPU. A hypothesis is that the level of CPU in the conditioned media is too low to be detected with the PATAFI-S antibody, while proCPU is above the limit of detection for this antibody and vice versa for the CP17 antibody.

By incubating HepG2, THP-1 and PMA-stimulated THP-1 cells with a specific CPU substrate and by detecting the cleaved substrate in the cell environment, it was demonstrated that proCPU can be activated into enzymatically active CPU in the in vitro cell culture environment. This confirmed that it was truly CPU that was detected on the CP17 Western blot. To further substantiate that the liver-derived CPU and monocyte/macrophage-derived CPU possessed the same peptidase activity in a biological context, studying the activity of monocyte/macrophage-derived CPU in a functional assay is of interest. This could be conducted by using an in vitro clot lysis assay in which monocyte/macrophage-derived CPU is added to the assay, and then comparing the results with those of an in vitro clot lysis experiment to which liver-derived proCPU is added.

In addition to Western blotting, proCPU was also measured in the concentrated conditioned media using an in-house enzymatic assay. In accordance with the results of the mRNA analysis, a decrease in proCPU concentration in the cell medium was found upon monocyte-to-macrophage differentiation, and this phenomenon was seen in the monocytic cell line as well as in the primary cells. In contrast to *CPB2* mRNA expression, the proCPU levels in the medium of THP-1 cells and primary human monocytes were very similar. A discrepancy between mRNA and protein abundance as observed for the THP-1/PMA-stimulated THP-1 cells was, however, frequently seen, and a theoretical understanding mostly remains elusive [27]. Aside from this, these findings support the hypothesis of Lin et al. that proCPU is expressed in monocytes and macrophages [16]. The presence of proCPU in the macrophages of atherosclerotic plaques, as detected by Rylander and co-workers, thus cannot solely be attributed to the phagocytosis of environmental proCPU by these macrophages [21]. Although their proCPU levels are low compared to the plasma proCPU concentrations, monocytes and macrophages may provide a local source of proCPU and boost proCPU concentrations, resulting in an additive effect on fibrinolysis driven by plasma proCPU [28,29]. The downregulation of proCPU levels in the cell medium during monocyte-to-macrophage differentiation suggests a more pronounced role for this enzyme in monocytes compared to macrophages. However, the exact role and significance of these cells as local proCPU sources are not clear at this time.

The incubation of primary M-CSF primed macrophages with IFN-ɣ/LPS had little influence on the proCPU levels, while alternative macrophage activation with IL-4 caused a further downregulation in the proCPU levels. The significance of the differential expression of *CPB2* mRNA and the secretion of the proCPU protein by classically and alternatively activated macrophages is still also an open question. In the setting of atherosclerosis, IFN-γ/LPS-stimulated macrophages are associated with symptomatic and unstable plaques, whereas IL-4-stimulated macrophages are particularly abundant in stable zones of the plaque and asymptomatic lesions [2,10,11,21]. In addition, it was recently shown that proCPU/CPU is present in considerable amounts in carotid plaques, with the highest levels corresponding to the vulnerable part of the plaque, adjacent to an area with high macrophage/foam cell content and substantial neovascularization [21]. Based on this, we speculate that there might be a need for plaque stabilizing mechanisms in unstable, M1-rich plaques. Since CPU limits plasmin generation, thereby preventing fragmentation of the fibrin network (fibrinolysis), counteracting destabilizing effects in this environment and contributing to keeping the plaque intact, it seems plausible that the presence of higher proCPU concentrations in this environment (through proCPU expression by IFN-γ/LPS-stimulated macrophages) is one such mechanism. Following this hypothesis, it seems logical that proCPU expression is the lowest in the IL-4-stimulated macrophages. These type of macrophages are predominantly present in the stable environment of the plaque, where there is little or no need for additional plaque stabilizing mechanisms.

## 4. Materials and Methods

### 4.1. Cell Culture

#### 4.1.1. Cell Lines

HepG2 cells (human hepatocellular carcinoma; Sigma-Aldrich, Saint-Louis, MO, USA) and THP-1 cells (human acute monocytic leukemia, ATCC) were respectively grown in DMEM or RPMI 1640 both supplemented with 10% fetal calf serum (FCS), 100 U/mL penicillin, and 100 µg/mL streptomycin. Part of the THP-1 cells was differentiated into a macrophage-like phenotype by the addition of 0.2 μM phorbol 12-myristate 13-acetate (PMA; Sigma-Aldrich) to the medium for 72 h. All cells were incubated at 37 °C under a 95% air/5% CO_2_ atmosphere. Passage numbers 2–5 were used for HepG2 and 2–8 for the THP-1 cells.

#### 4.1.2. Primary Cells

Human PBMCs were isolated from buffy coats of anonymous clinically healthy blood donors (Red Cross, Mechelen, Belgium) by Ficoll–Paque Premium gradient centrifugation. Ethical approval for the buffy coats and processes used in this study was given by the Ethics Committee UZA/UAntwerp (B300201939437) and all donors (N = 17) gave their written informed consent. Briefly, a 40 mL buffy coat was diluted in PBS (1:1, *v*/*v*), layered on top of Ficoll–Paque Premium solution (GE Healthcare, Machelen, Belgium) and centrifuged (40 min, 400× *g*, no brakes). PBMCs were collected from the interface and washed twice with PBS. CD14+ monocytes were enriched from the freshly isolated mononuclear cell fraction via CD14+ positive magnetic selection using CD14-microbeads (20 µL of microbeads per 10^7^ total PBMCs; Miltenyi Biotec, Bergisch Gladbach, Germany) following the manufacturer’s protocol. MACS-purified CD14+ monocytes were then seeded at a density of 2 × 10^6^ cells/mL in complete RPMI medium and placed in a humidified incubator with 5% CO_2_ at 37 °C. Monocytes were either harvested after 24 h of culturing or differentiated into macrophages immediately after seeding.

For monocyte-to-macrophage differentiation, freshly isolated human CD14+ monocytes were seeded at a density of 2 × 10^6^ cells/mL in complete medium supplemented with 20 ng/mL recombinant human macrophage colony-stimulating factor (rhM-CSF, Immunotools, Friesoythe, Germany) [30]. After 5 days of incubation, macrophages were harvested or further polarized by 2 days of incubation with 20 ng/mL rhM-CSF in combination with either 100 U/mL IFN-γ (Immunotools) and 20 ng/mL LPS (Immunotools) to obtain classically activated macrophages or 20 ng/mL IL-4 (Immuntools) for alternatively activated macrophages [30,31]. To confirm appropriate macrophage polarization by these stimulation protocols, the medium was aspirated 8 and 24 h after the start of the polarization [31]. TNF-α levels were determined in the 8 h aspirate and the IL-6, IL-1β and IL-10 levels in the 24 h aspirate using the respective ELISA (hTNF-α ELISA, hIL-6 ELISA, hIL-1β ELISA, and hIL-10 ELISA; Immunotools) (Appendix A).

### 4.2. Conditioned Media and Cellular Lysates

Conditioned medium of the different cell types was obtained by replacing complete medium with 5% of the respective complete medium and 95% Hank’s Balanced Salt Solution (HBSS, Gibco, Waltham, MA, USA) 24 h before harvesting. After 24 h, the conditioned medium was collected and stored at −80 °C until further analysis. Cellular lysates were prepared by washing 2 × 10^6^ cells twice with PBS and suspending these cells in 50 µL of the appropriate lysis buffer: lysis buffer for proCPU measurement (1% octylglucoside, 10 mM EDTA, 70 µg/mL aprotinin, 50 mM Tris-HCl pH 8.3) or lysis buffer for Western blot analysis (1% Triton X-100, 150 mM NaCl, Complete Protease Inhibitor Cocktail (Roche Diagnostics, Brussels, Belgium), 50 mM Tris pH 7.6). After 1 h on ice with frequent agitation, the samples were centrifuged at 12,000× *g* for 10 min at 4 °C and the cellular lysate was collected. Conditioned media and cellular lysates were stored at −80 °C. The protein content of the conditioned media and cellular lysates was determined according to the Bradford method using bovine serum albumin (Sigma-Aldrich) as a standard [32].

### 4.3. RNA Isolation and cDNA Synthesis for mRNA Expression Analysis

Total RNA was isolated from 2 × 10^6^ cells using the SV Total RNA Isolation System Kit (Promega, Madison, WI, USA) following the manufacturer’s instructions. RNA quality and concentration were assessed by measuring the absorbance at 230, 260, and 280 nm using a UV–Visible spectrophotometer (Nanodrop 2000, Thermo Fisher Scientific, Waltham, MA, USA). Next, first-strand cDNA was synthesized starting from 2 µg of the extracted total RNA and using the Omniscript^®^ Reverse Transcription Kit (Qiagen, Hilden, Duitsland).

### 4.4. CPB2 mRNA Expression in Human Monocyte and Macrophage Cell Lines and Primary Cells

Prior to the measurement of *CPB2* mRNA expression in the different cell types, a quantitative reverse transcriptase-polymerase chain reaction (RT-qPCR) assay to study CPB2 mRNA expression was validated by applying the MIQE guidelines [33] (Appendix A) and appropriate reference genes for relative quantification were selected (Appendix A). Subsequently, Cq values of the gene of interest, CPB2, were determined by RT-qPCR in all of the investigated cell types and *CPB2* mRNA expression was expressed relative to the previously selected reference genes [34].

### 4.5. Western Blot Analysis

Purified proCPU was obtained from human plasma as previously described [35]. Prior to Western blot analysis, the conditioned medium of at least four independent experiments/donors was pooled for each cell type and concentrated 20-fold using 10 K centrifugal filter devices (Amicon^®^ Ultra-0.5, Merck Millipore, Burlington, MA, USA). To obtain active CPU as a control, purified proCPU was activated at 25 °C with thrombin-thrombomodulin (4 nM and 16 nM, respectively) in the presence of 50 mM CaCl_2_. The reaction was stopped after 20 min by adding 4X sample buffer and immediately boiling the samples. All samples were then subjected to Western blot analysis. Following electrophoresis on a 10% SDS-PAGE gel, the proteins were transferred to a nitrocellulose membrane by electroblotting. Blocking of non-specific binding sites was achieved with 5% BSA in washing buffer [0.05 M Tris, 0.15 M NaCl, 0.15% Tween 20, pH 7.4] for 1 h at room temperature. Next, blots were incubated overnight with primary antibodies diluted in blocking buffer: polyclonal sheep anti-human proCPU antibody (PATAFI-S, Prolytix, Essex Junction, VT, USA; 1:1500) or polyclonal rabbit anti-human proCPU/CPU antibody (CP17, Agrisera, Vännäs, Sweden; 1:1500). Subsequently, secondary antibodies diluted in blocking buffer were added for 2 h at room temperature: goat anti-sheep horseradish peroxidase (HRP) (31480, ThermoFisher; 1:5000) and goat anti-rabbit HRP (65-6120, Invitrogen; 1:5000) were used. Between the different incubations, membranes were washed 6 × 5 min with washing buffer. Chemiluminescent detection was performed using the SuperSignal West Femto Substrate Kit (Thermo Fisher Scientific). The protein bands were visualized on a OptiGo viewer and Proxima AQ-4 software. Precision Plus Protein Dual Color Standards (Bio-Rad, Hercules, CA, USA) were used for MW estimation.

### 4.6. ProCPU Measurement

ProCPU concentrations were measured in both the conditioned media and cellular lysate using a previously described, in-house enzymatic assay [36] with the modification that the conditioned media samples and the cellular lysates were concentrated 3- to 4-fold prior to proCPU measurement using a 10K centrifugal filter device (Amicon^®^ Ultra-0.5, Merck Millipore). Samples were then incubated with AZD9684 (a potent and selective small-molecule CPU inhibitor that was a kind gift from AstraZeneca; final concentration 5 µM) or an equal volume of HEPES (20 mmol/L; pH 7.4) for 5 min [37,38]. Hereafter a mixture of human thrombin (Merck), rabbit-lung thrombomodulin (Seikisui Diagnostics, Burlington, MA, USA) and CaCl_2_ (Merck) (final concentrations of 4 nM, 16 nM, and 50 mM, respectively) was added to quantitatively convert proCPU into the active enzyme. Subsequently, the active CPU was incubated with the selective and specific substrate Bz-*o*-cyano-Phe-Arg, followed by quantification of the formed product by high-performance liquid chromatography. The enzymatic activity measured in the presence of AZD9684 was then subtracted from the enzymatic activity in the absence of AZD9684 to obtain the actual proCPU concentration and to exclude that the measured activity originated from other basis carboxypeptidases or other enzymes that could be present in the samples [36].

### 4.7. Statistical Analysis

Statistics were performed using IBM SPSS Statistics 27 and figures were compiled in GraphPad Prism 9.3.1. The specific statistical tests used in this study are mentioned in the legends underneath the figures. Data are presented as the mean ± standard error of the mean (SEM). Results were considered significant if the *p*-value was < 0.05.

## 5. Conclusions

In this study, we confirmed the expression of *CPB2* mRNA by THP-1 and PMA-stimulated THP-1 cells and showed that *CPB2* mRNA is expressed in primary human monocytes isolated from PBMCs and macrophages derived from these monocytes. On a protein level, proCPU was detected in the conditioned medium of all of the investigated cell types. Moreover, it was demonstrated that proCPU can be activated into functionally active CPU in the in vitro cell culture environment. Comparison of both the relative *CPB2* mRNA expression and proCPU concentrations in the cell medium between the different cell types provide evidence that *CPB2* mRNA expression and proCPU secretion in the monocytes and (activated) macrophages are related to the degree to which these cells are differentiated and activated. This sheds new light on monocytes and macrophages as local proCPU sources within atherosclerotic plaques and extra-vascular inflammatory sites and the potential role of the proCPU system as a modulator of inflammation in these environments.

## Figures and Tables

**Figure 1 ijms-24-03725-f001:**
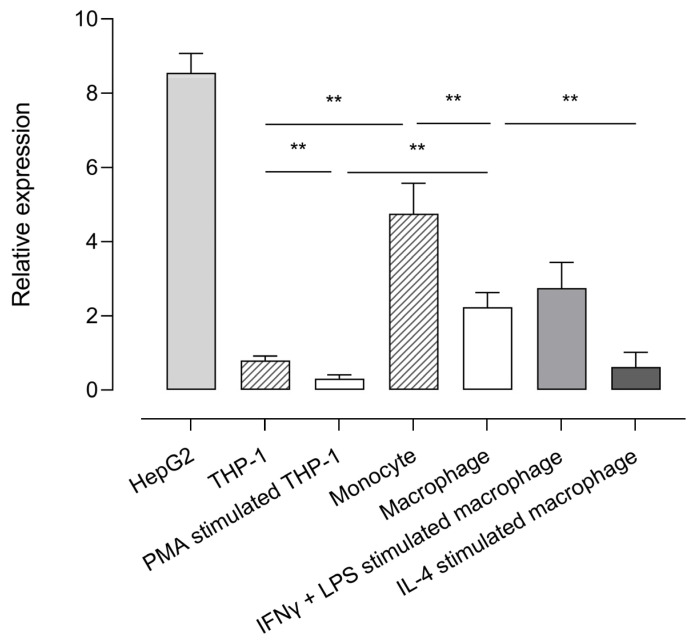
Relative expression of *CPB2* mRNA in various cell types. The abundance of *CPB2* mRNA in HepG2, THP-1, PMA-stimulated THP-1 cells (macrophage-like phenotype), primary human monocytes, primary human M-CSF primed macrophages, IFN-ɣ/LPS-stimulated macrophages (M1 macrophage, classically activated), and IL-4-stimulated macrophages (M2 macrophage, alternatively activated) was analyzed by quantitative reverse-transcriptase polymerase chain reaction (RT-qPCR). *CPB2* mRNA expression was displayed relative to a set of stable reference genes (ARCP1a, EMC7, and TBP). Data present the mean of 5–7 biological replicates measured in triplicate. Error bars represent the standard error of the mean (SEM). A Kruskal–Wallis test with Dunn’s multiple comparison test was performed to test for statistical significance between all groups of monocytes/macrophages. Statistically significant differences are indicated with an asterisk (** *p* < 0.01).

**Figure 2 ijms-24-03725-f002:**
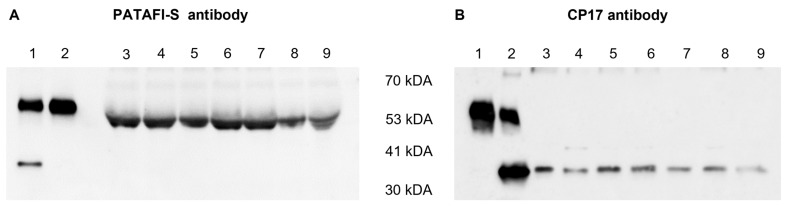
Detection of human proCPU and CPU protein in the concentrated conditioned media of various cell types. (**A**) Western blot using a polyclonal sheep anti-human proCPU antibody (PATAF-S, Prolytix). Lane 1: Mixture of purified plasma proCPU (58 kDa) and CPU (35 kDa) as the control, Lane 2: Purified plasma proCPU as the control, Lanes 3–9: Concentrated conditioned media HepG2 (lane 3), THP-1 (lane 4), PMA-stimulated THP-1 (lane 5), primary human monocytes (lane 6), primary human M-CSF macrophage (lane 7), primary human IFN-ɣ/LPS-stimulated macrophage (lane 8), primary human IL-4-stimulated macrophage (lane 9). (**B**) Western blot using a polyclonal rabbit anti-human proCPU/CPU antibody (CP17, Agrisera). Lane 1: Purified plasma proCPU as the control, Lane 2: Mixture of purified plasma proCPU (58 kDa) and CPU (35 kDa) as the control, Lanes 3–9: Concentrated conditioned media: HepG2 (lane 3), THP-1 (lane 4), PMA-stimulated THP-1 (lane 5), primary human monocytes (lane 6), primary human M-CSF macrophage (lane 7), primary human IFN-ɣ/LPS-stimulated macrophage (lane 8), primary human IL-4-stimulated macrophage (lane 9).

**Figure 3 ijms-24-03725-f003:**
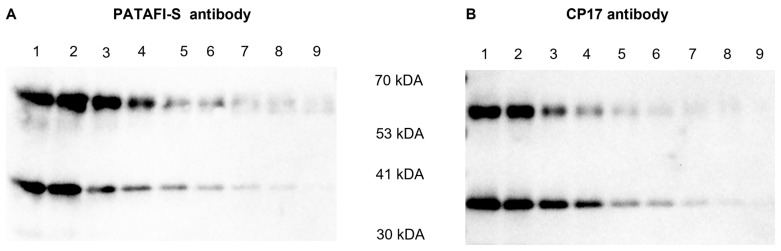
Reactivity of a polyclonal sheep anti-human proCPU antibody (PATAFI-S) (**A**) and a polyclonal rabbit anti-human proCPU/CPU antibody (CP17) (**B**) toward proCPU and CPU. A serial dilution of a mixture of purified plasma proCPU (58 kDa) and CPU (36 kDa) was prepared and subjected to SDS-PAGE and Western blotting (PATAFI: lanes 1–9; CP17: lanes 1–9) to gain insights into the reactivity of both polyclonal antibodies against proCPU and/or CPU.

**Figure 4 ijms-24-03725-f004:**
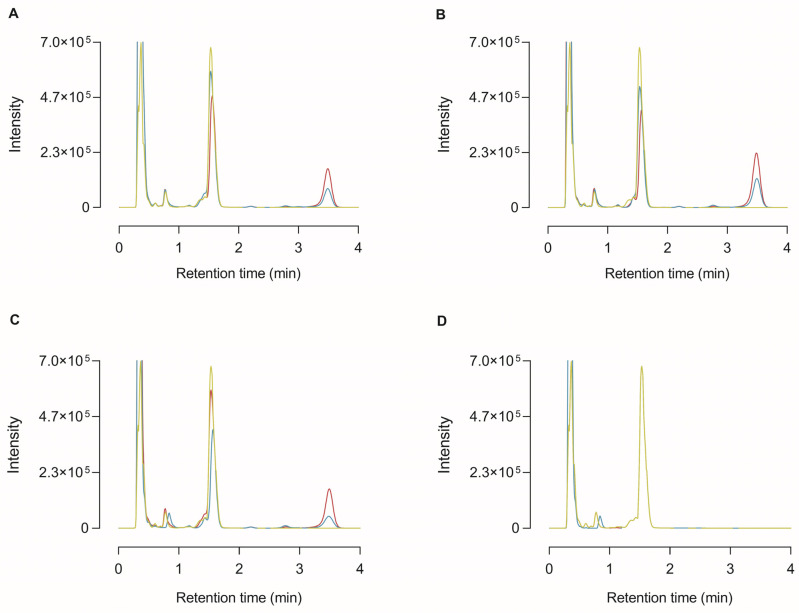
High-performance liquid chromatography (HPLC) chromatogram of medium samples from the cleavage of the selective CPU substrate Bz-*o*-cyano-Phe-Arg (final concentration 1 mM) by HepG2 (**A**), THP-1 (**B**), PMA-stimulated THP-1 cells (**C**), or in the absence of any cells (blanc; (**D**)). At 0 h (green curve), 24 h (blue curve), and 72 h (red curve) after incubation with the substrate, the cell medium was collected and analyzed by reversed-phase HPLC. The retention time of Bz-*o*-cyano-Phe-Arg (substrate) and Bz-*o*-cyano-Phe (cleaved substrate) was ±1.53 min and ±3.49 min, respectively.

**Figure 5 ijms-24-03725-f005:**
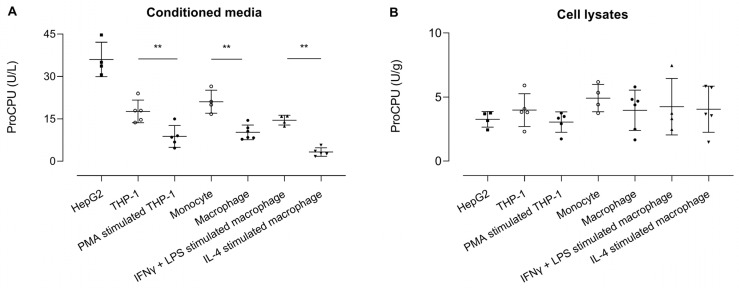
ProCPU activity levels measured in (**A**) concentrated conditioned media (U/L) and (**B**) concentrated cell lysates (U/g) of various cell types. Results are reported as the mean ± SEM (N = 4–6; for primary cells, each data point represents the result from an independent experiment with cells obtained from another donor for each experiment). A Kruskal–Wallis test with Dunn’s multiple comparison test was used to test for statistical significance between all groups; ** *p* < 0.01.

## Data Availability

Data are contained within the article or Appendix A.

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
