# Peer review of "ProCPU Is Expressed by (Primary) Human Monocytes and Macrophages and Expression Differs between States of Differentiation and Activation"

_ijms, 2023, doi:10.3390/ijms24043725_

Round 1

Reviewer 1 Report

This study indicates, besides synthesized by liver, proCPU can be expressed by primary monocytes and macrophages, and it varies between states of differentiation and activation. These novel findings provide new light on monocytes and macrophages as local proCPU sources, which can contribute to the mechanism study of thrombus formation. Yet the week point of this study includes the evidence provided to support the hypothesis is insufficient. More experiments are suggested make this conclusion. Taken together, I agree to publish this work after major revisions are made and extra experiments are conducted. Please consider the comments listed as below.

1.      Blood samples from the inflammatory patients especially patients with thrombus are strongly suggested to be included as a positive control. ProCPU expression level on induced patient monocytes and macrophages are expected to be elevated compared to the healthy human samples.

2.      Could you provide any in vitro or in vivo evidence proving that the proCPU expressed by monocytes and macrophages possesses the same function as the proCPU synthesized by the liver, for instance, inducing coagulation and/or inflammation?

3.      Can the proCPU antagonist and/or blocking antibody inhibit its expression on monocytes and macrophages?

Reviewer 2 Report

The authors have shown that proCPU is expressed in primary macrophages and monocytes and here is where the novelty of the study lies. They proved this expression both at the transcript and protein levels. They also show that the degree of differentiation is related to the expression levels. 

- More literature and extension of the introduction would be suggested as the novelty and the importance of the findings is not highlighted enough.

- How do the authors explain the failure to detect proCPU/CPU in lysates?

- The western blot of Figure S7 should be repeated. The quality is poor.
